# On the recent contribution of the Greenland ice sheet to sea level change

Michiel van den Broeke<sup>1</sup>, Ellyn Enderlin<sup>2</sup>, Ian Howat<sup>3</sup>, Peter Kuipers Munneke<sup>1</sup>, Brice Noël<sup>1</sup>, Willem Jan van de Berg<sup>1</sup>, Erik van Meijgaard<sup>4</sup>, and Bert Wouters<sup>1</sup>

<sup>1</sup>Institute for Marine and Atmospheric Research Utrecht, Utrecht University, The Netherlands <sup>2</sup>Climate Change Institute and School of Earth and Climate Sciences, University of Maine, USA <sup>3</sup>School of Earth Sciences, Ohio State University, USA <sup>4</sup>Royal Netherlands Meteorological Institute, De Bilt, The Netherlands

Correspondence to: Michiel van den Broeke (m.r.vandenbroeke@uu.nl)

Abstract. We assess the recent contribution of the Greenland ice sheet (GrIS) to sea level change. We use the mass budget method, which quantifies ice sheet mass balance (MB) as the difference between surface mass balance (SMB) and solid ice discharge across the grounding line (D). A comparison with independent gravity change observations from GRACE shows good agreement for the overlapping period 2002-2015, giving confidence in the partitioning of recent GrIS mass changes.

- 5 The estimated 1995 value of D and the 1958-1995 average value of SMB are similar at 411 and 418 Gt yr<sup>-1</sup>, respectively, suggesting that ice flow in the mid-nineties was well adjusted to the average annual mass input, reminiscent of an ice sheet in approximate balance. Starting in the early to mid-1990's, SMB decreased while D increased, leading to quasi-persistent negative MB. About 60% of the associated mass loss since 1991 is caused by changes in SMB and the remainder by D. The decrease in SMB is fully driven by an increase in surface melt and subsequent meltwater runoff, which is slightly compensated
- by a small (< 3%) increase in snowfall. The excess runoff originates from low-lying (< 2000 m a.s.l.) parts of the ice sheet; higher up, increased refreezing prevents runoff of meltwater from occurring, at the expense of increased firn temperatures and depleted pore space. With a 1991-2015 average annual mass loss of  $\sim$ 0.47  $\pm$  0.23 mm sea level equivalent (SLE) and a peak contribution of 1.2 mm SLE in 2012, the GrIS has recently become a major source of global mean sea level rise.

### 1 Introduction

Three methods are commonly used to assess the mass balance of the Greenland ice sheet (GrIS): gravimetry, radar/laser altimetry and the mass budget method. Each method has distinct advantages and disadvantages. The main advantage of gravity observations is that they provide, once corrected and deconvolved, a relatively direct measure of glacial mass change; as a

- 5 result, several years after its launch, the dual-satellite Gravity Recovery and Climate Experiment (GRACE) mission was able to confirm that the GrIS is losing mass (Velicogna and Wahr, 2006). The main drawbacks of this technique are the relative brevity of the time series (since late 2002), the large footprint of ~200 km and the fact that corrections must be made for mass movements in atmosphere, ocean, soil and solid earth (e.g. glacial isostatic adjustment, or GIA) in order to isolate ice mass changes.
- Like GRACE, satellite altimetry provides full coverage of the GrIS. The time series are longer, starting in the mid-1990's, with aerial photography extending some of the records back to the mid-1980's and earlier (Kjær et al., 2012; Kjeldsen et al., 2015). Difficulties in interpreting radar altimeter data arise from the variable penetration depth of the radar signal in firm (Thomas et al., 2008) and, especially for the earlier instruments, signal loss along the steep coastal margins. The radar altimeter onboard CryoSat-2, launched in 2010, partly mitigates these issues and shows GrIS elevation changes in unprecedented detail
- 15 (Helm et al., 2014). ICESat's laser altimeter measured the surface elevation change accurately but was sensitive to cloud cover and had a relatively large ground track separation. The various altimeter records must be inter-calibrated and spatially interpolated between ground tracks before a continuous time series is obtained. Both radar and laser altimeters measure ice sheet volume changes which must be converted to mass changes using a model that accounts for vertical bedrock motion and variability in the depth and mass of the firn layer, which introduces additional uncertainties (Kuipers Munneke et al., 2015).
- The mass budget method (MBM) estimates the difference between individual mass sources (mainly snowfall) and sinks (mainly meltwater runoff and solid ice discharge). Because it resolves the individual components of the mass balance, this method has the advantage that it identifies the physical processes that are responsible for the mass change. However, because the mass change represents the relatively small difference between three large source and sink terms, it is very sensitive to uncertainties in any of these. This is especially true for surface mass fluxes such as snowfall and meltwater runoff; because
- 25 these cannot be measured from space, they must be interpolated from scarce *in situ* measurements and/or simulated using dedicated regional climate models, which introduces potentially large uncertainties (Vernon et al., 2013).

Shepherd et al. (2012) reconciled results of the three methods for the GrIS to obtain an average GrIS 1992-2011 mass loss of  $142 \pm 49$  Gt yr<sup>-1</sup>. The fifth assessment report of Working Group I of the Intergovernmental Panel on Climate Change (IPCC), while weighing the results of the various studies somewhat differently, arrived at a similar conclusion and shows that in 2012

30

the GrIS had become the largest single contributor to sea level rise (Vaughan et al., 2013). A compilation of GrIS mass balance studies covering sub-periods of the satellite era confirms that the mass loss of the GrIS is accelerating (Hanna et al., 2013b).

Combining two or more methods may reduce uncertainties and may provide additional insights in the physical processes causing the mass loss. Sasgen et al. (2012) used GRACE, the MBM and altimetry to assess mass changes in seven GrIS regions; for the GrIS as a whole, between August 2002 and August 2010 they found a mass trend of  $228 \pm 22$  Gt yr<sup>-1</sup>, with

an acceleration of  $15 \pm 7$  Gt yr<sup>-2</sup>. Rignot et al. (2011) used a combination of GRACE and the MBM to show that GrIS mass loss between 1992 and 2010 accelerated by 17-22 Gt yr<sup>-2</sup>. When combined with surface mass balance fields, satellite altimetry can be used to spatially separate mass losses arising from ice dynamical (i.e. driven by ice flow) and surface (i.e. driven by the atmosphere) processes (Csatho et al., 2014; Kjeldsen et al., 2015). Using anomalies relative to a reference period, Van den

5 Broeke et al. (2009) showed that 1996-2008 GrIS mass loss was approximately equally partitioned between increased surface meltwater runoff and ice discharge. Enderlin et al. (2014) showed that the relative contribution of ice discharge to total GrIS mass loss decreased from 58% before 2005 to 32% between 2009 and 2012, indicating an increasingly important role for surface processes.

In this paper we combine GRACE data and the MBM, using new *SMB* and *D* data that allow updating the time series 10 to 2015, to identify the causes and temporal evolution of recent (1958-2015) GrIS mass loss and its contribution to sea level rise. In Section 2 we discuss methods, in Section 3 we discuss results and in Section 4 we identify outstanding problems and avenues for future research.

# 2 Methods

25

#### 2.1 Definitions

15 The mass balance (MB) of an ice sheet, usually expressed in Gt yr<sup>-1</sup>, represents the change of its mass in time dM/dt. Neglecting basal melting of grounded ice, which typically does not exceed several mm per year, and assuming the grounding line position to remain unchanged, the MB of the grounded ice sheet is governed by the difference between surface mass balance (SMB) and solid ice discharge across the grounding line (D):

$$MB = \frac{dM}{dt} = SMB - D \tag{1}$$

20 *SMB* represents the sum of mass fluxes towards and away from the ice sheet surface:

$$SMB = P_{tot} - SU_{tot} - ER_{ds} - RU \tag{2}$$

where  $P_{tot}$  is total precipitative flux (sum of snowfall SN and rainfall RA),  $SU_{tot}$  is total sublimation (from the surface and from drifting snow particles),  $ER_{ds}$  is erosion of snow by divergence of the drifting snow transport and RU is meltwater runoff.<sup>1</sup> The accumulation/ablation zones of an ice sheet are defined as the areas where SMB > 0 and SMB < 0, respectively. These two zones are separated by the equilibrium line, where SMB = 0.

 $<sup>^{1}</sup>SMB$  as defined here, i.e. including the subsurface processes retention and refreezing, is sometimes also referred to as *climatic mass balance*, to distinguish it from processes taking place purely at the surface.

The amount of runoff RU from the ice sheet is determined by the liquid water balance LWB, the sum of all sources (mainly rainfall and melt) and sinks (mainly refreezing and capillary retention) of liquid water in the column of firn and/or ice:

$$RU = RA + CO + ME - RT - RF \tag{3}$$

where CO is condensation of water vapour at the ice sheet surface, ME is surface meltwater production, RT is retention of 5 liquid water in the snow/firn by capillary forces and RF is refreezing of liquid water at or below the surface. Equation (3) in first order includes processes associated with the formation of perennial firn aquifers (Forster et al., 2013; Kuipers Munneke et al., 2014), but neglects the delay in runoff by storage of meltwater in semi-permanent supra-, sub- and englacial lakes and channels, which is potentially significant (Rennermalm et al., 2013).

In summary, to quantify the mass balance of the GriS, the MBM relies on the quantification of mass sinks and sources in 10 equations (1), (2) and (3).

#### 2.2 Data sources

To calculate GrIS MB using the mass budget method we combine SMB components calculated with the regional atmospheric climate model RACMO2.3 with annual estimates of D, updated from Enderlin et al. (2014). The latter data represent discharge summed over all marine terminating glaciers wider than 1 km, and cover the 16-year period 2000-2015. Between 1996 and

- 2000 we assume a linear increase in D of in total 38 Gt yr<sup>-1</sup> following Rignot et al. (2008). In the absence of data, no changes in D are assumed between 1958 and 1996. A seasonal cycle in ice sheet wide D is not considered, although it is well known that marine-terminating outlet glaciers can show pronounced (sub-)seasonal velocity oscillations (Moon et al., 2014). Because seasonal acceleration leads to thinning, the effect on D is smaller than based on velocity changes alone. Moreover, by using the median annual velocity, seasonal variability is accounted for to a large extent (Enderlin et al., 2014). Land-terminating glaciers
- also exhibit a seasonal cycle in their velocity (Van de Wal et al., 2008; Joughin et al., 2008; Bartholomew et al., 2011; Sole et al., 2013), but this does not influence *D*.

SMB components are derived from a run with RACMO2.3 for the period January 1958-December 2015, using 40 vertical layers and a horizontal resolution of ~11 x 11 km<sup>2</sup> (Nöel et al., 2015). Figure 1 shows the Greenland domain of RACMO2.3, which consists of 312 (latitude) x 306 (longitude) grid cells and includes Iceland, the Svalbard archipelago and the Canadian

- Arctic. The model is forced at the lateral boundaries by the 40-year European Center for Medium-Range Weather Forecasts (ECMWF) Re-Analysis (ERA-40) for the period January 1958 - December 1979 and the ECMWF Interim Reanalysis (ERA-Interim) afterwards. In previous versions of RACMO2, the impact of using inhomogeneous forcing before and after 1979 (ERA-40 vs ERA-Int) was found to be small (Ettema et al., 2009). According to Fettweis et al. (2013a), the regional climate model MAR shows precipitation to be 5% greater when forced by ERA-40 compared to ERA-Interim for the period 1980-
- 1999, probably a result of biases in the ERA-40 humidity scheme that were corrected in ERA-Interim. This uncertainty falls within the error bar used here.

The polar version of RACMO2.3 has been especially developed to simulate SMB of glaciated regions and is an update of RACMO2.1 (Ettema et al., 2009; Van Angelen et al., 2014). It is interactively coupled to a multilayer one-dimensional (column) firn model that simulates (sub-)surface processes like vertical heat transport, grain growth, firn densification, meltwater percolation, retention and refreezing. RACMO2.3 uses a prognostic calculation of snow grain size from which broadband snow

- 5 albedo is derived (Kuipers Munneke et al., 2011). Ice albedo is not explicitly calculated, as it is influenced by poorly known processes such as dust accumulation and biological activity; to account for its considerable spatial heterogeneity (Bøggild et al., 2010), ice albedo is prescribed from the Moderate Resolution Imaging Spectroradiometer (MODIS) onboard the Terra and Aqua satellites (Stroeve et al., 2005).
- RACMO2.3 includes a routine for drifting snow sublimation, which removes on average 25 Gt yr<sup>-1</sup> from the ice sheet
  (Lenaerts et al., 2012), with little interannual variability; when compared to scarce observations, this scheme was found to accurately predict the occurrence of drifting snow, but to overestimate drifting snow transport, and thus likely also overestimate drifting snow sublimation, although no direct observations are available to verify this. For a more detailed description of recent changes in RACMO2.3 model physics and how that impacts the modelled *SMB* of the GrIS, the reader is referred to Nöel et al. (2015) and references therein.
- Other RACMO2.3 modelled *SMB* components and atmospheric parameters have been extensively evaluated against *in situ* observations both in the accumulation and ablation zone of the GrIS (Ettema et al., 2009; Nöel et al., 2015). From these comparisons, typical uncertainties of 9% and 15 % were found for ice-sheet wide integrated accumulation and ablation, respectively. These are combined into an uncertainty for ice sheet wide *SMB* assuming accumulation and ablation to be independent. We note that this assumption is debatable, as ablation and accumulation tend to be connected via surface albedo, especially
- in summer (Fyke et al., 2014; Nöel et al., 2015). The scarcity of accumulation and ablation measurements does not allow for a further regionalization of the uncertainty, but obviously uncertainties can be significantly larger for smaller areas and subclimatological time periods. For the trend in cumulative SMB-D, an uncertainty is derived following Van den Broeke et al. (2009).
- We compare ice sheet mass balance obtained from the mass budget method, i.e. MB = SMB D, with monthly Gravity 25 Recovery And Climate Experiment (GRACE) gravity field solutions. Mass variations for the GrIS were derived from the CSR RL05 data release (April 2002- September 2015), following the method described in Wouters et al. (2008). In brief, regional mass anomalies are adjusted in a model consisting of eight pre-defined GrIS basins and the resulting gravity disturbance is computed. The modelled mass anomalies are then adjusted until convergence with the actual GRACE observations is reached. All standard corrections are applied to the GRACE data, including a correction for GIA, based on the model of A et al. (2013).
- The uncertainties in monthly GRACE values and mass trend are estimated at 40 Gt and 20 Gt yr<sup>-1</sup>, respectively. For the monthly values, the uncertainty is computed by conservatively assuming that all observed signals with a periodicity smaller than 6 months are due to noise (Wahr et al., 2006). For the trends, the quoted uncertainty takes into account methodological differences in the processing of the level-1 GRACE data, the uncertainty in the GIA correction, the formal error of the least-square fit and aliasing of ocean tides (Wouters et al., 2013). To assess the methodological uncertainty in the GRACE time

series, results were compared to mass anomalies from the Jet Propulsion Laboratory (JPL) mascons using a fully independent approach (Watkins et al., 2015). No significant differences in trend and interannual variability were found.

GRACE data are typically provided for mid-month, while cumulated SMB values from RACMO2.3 represent the end of the month, so we linearly interpolated the GRACE data to the end of the month. Missing monthly values were linearly interpolated.

Because GRACE provides mass anomalies in time  $\delta MB(t)$  rather than mass balance dM/dt, SMB and D must be integrated in time before being subtracted and compared with GRACE. Alternatively, we could also use the temporal derivative of the GRACE time series to obtain dM/dt, but given the inherent noise in the GRACE data this would introduce large uncertainties.

# 3 Results

# 3.1 Comparing MBM and GRACE

- In this paper we are principally interested in the mass balance of the contiguous ice sheet, the main reason being that Greenlands peripheral glaciers and ice caps (GIC) are usually assumed to be part of the global population of glaciers when a mass balance assessment is made, e.g. Radic and Hock (2011). However, because GRACE has a footprint similar to the maximum width of the ice-free tundra in Greenland (~200 km), it cannot readily separate mass changes of the contiguous ice sheet from those of GIC. Moreover, GRACE mass signals also include the waxing and waning of the tundra seasonal snow cover and hydrological
- signals. Although the latter two processes do in principle not contribute to long-term mass changes, because of their seasonal character, they do represent a significant seasonal cycle in mass loading over Greenland that varies from year to year, modifying the amplitude of the GRACE signal (Bevis et al., 2012). To enable a meaningful comparison with GRACE therefore requires integrating surface mass fluxes over the entire island, including the GIC and ice-free tundra. This is rather straightforward, because both are explicitly modelled in RACMO2.3, albeit the seasonal snow cover with a simpler (single-layer, no refreezing)
- snow model (Dutra et al., 2010). This island-integrated SMB, only used in this subsection (3.1), is indicated by  $SMB_{\text{Greenland}}$ . No marine terminating outlet glaciers wider than 1 km wide originate from detached ice caps, so we neglect their contribution to D and assume mass changes in GIC to be solely caused by SMB, i.e.  $D_{\text{Greenland}} = D$  and  $MB_{\text{Greenland}} = SMB_{\text{Greenland}} - D$ . The most direct way to compare MBM and GRACE is to cumulate  $SMB_{\text{Greenland}}$  and D in time and calculate the difference to get cumulative  $MB_{\text{Greenland}}$ , i.e.  $\delta MB$  relative to 1 January 1958. Figure 2 shows these cumulative values (expressed in Gt),
- starting at zero on 1 January 1958 when our time series of  $SMB_{Greenland}$  starts. For reference, the equivalent mass required for 5 cm global mean sea level change is indicated on the left. Until mid-1996, cumulative *D* represents a straight line, because its annual value is assumed constant at the 1996 value of 411 Gt yr<sup>-1</sup>. The average value of  $SMB_{Greenland}$  over the period 1958-1995 is within 2% of this value (418 Gt yr<sup>-1</sup>), resulting in an estimated pre-1996 cumulative mass balance (red line) that remains close to zero, in line with previous results of Howat and Eddy (2011). The fact that the estimated 1995 value of *D* and
- the 1958-1995 average value of SMB are similar suggests that ice flow in the mid-nineties was well adjusted to the average annual mass input of the previous decades, reminiscent of an ice sheet in approximate balance (Hanna et al., 2013b). Because we do not include a seasonal cycle in D, the mass curve shows a gradual wintertime increase, when  $SMB_{\text{Greenland}}$  exceeds D

in magnitude and Greenland gains mass, and a steep summer drop, when  $SMB_{Greenland}$  becomes strongly negative and acts together with D to remove mass from Greenland.

After 1995, following the acceleration of several large outlet glaciers in the southeast and northwest (Rignot and Kanagarathnam, 2006; Moon et al., 2012), D increased while simultaneously  $SMB_{\text{Greenland}}$  decreased. As a result, their cumulated

- values in Figure 2 curve upward and downward, respectively, and cumulative  $MB_{\text{Greenland}}$  becomes persistently negative as a 5 result. According to Figure 2, the most significant mass loss derives from the last 1-2 decades, and it is a fortunate coincidence that the GRACE mission covered most of this period. The recent Greenland mass evolution from the MBM agrees qualitatively well with the GRACE observations, represented by the dark grey line in Figure 2. Note that, because GRACE measures mass changes rather than absolute mass, the time series have been vertically offset by 1000 Gt for clarity without losing information.
- Figure 3a zooms in on the time series of cumulative  $MB_{\text{Greenland}}$  and GRACE during the overlapping period (2002-2015). 10 Although the seasonal oscillations in the GRACE time series have lost some of their amplitude because of interpolation, Figure 3b confirms the good agreement ( $R^2 > 0.995$ ) between the fully independent time series. The seasonal and interannual variations in the GRACE time series are qualitatively well reproduced, with the largest summertime mass losses in 2007, 2010 and 2012, limited mass loss in 2013 and large interannual variability in wintertime accumulation. The magnitude of a fitted
- linear trend over the period 2003-2014 is also similar,  $-294 \pm 5$  Gt yr<sup>-1</sup> in  $MB_{\text{Greenland}}$  and  $-270 \pm 4$  Gt yr<sup>-1</sup> in GRACE. These 15 errors represent fitting uncertainties; the real uncertainties in the trends are estimated at 20 Gt yr<sup>-1</sup> in both time series (Van den Broeke et al., 2009). This good agreement between both methods is partly fortuitous owing to compensating biases in distinct SMB components (see section 4). Nonetheless, Figures 2 and 3 inspire sufficient confidence to support a more quantitative analysis of the components of year-to-year contributions of the GrIS to global average sea level rise.

#### 20 3.2 **Temporal SMB variability**

In order to combine SMB with annual D in subsection 3.4 we integrated SMB components over the contiguous GrIS ice mask of RACMO2.3 and over calendar years. Figure 4 shows the resulting time series of the main SMB components, with 1991-2015 trends included as dashed lines. Table 1 lists the averages and trends over the climate period 1961-1990, for which the GrIS was assumed to be in approximate balance, and the recent melting period 1991-2015.

- 25 Total annual precipitation ( $P_{tot}$ ) typically varies between 600 and 800 Gt yr<sup>-1</sup>, without significant trend over the full period. A small positive trend in P<sub>tot</sub> of 0.3% per year for the period 1961-1990 is followed by an insignificant negative trend in the subsequent decades. Although total precipitation on the GrIS has not significantly changed over the last six decades, the rainfall RA fraction has increased in response to a warmer atmosphere. During 1961-1990, 3.3% or 23 Gt yr<sup>-1</sup> of the modelled precipitation on the ice sheet fell as rain, increasing to 3.9% or 28 Gt yr<sup>-1</sup> in 1991-2015. For the full period, the annual rain fraction varied between 2.0% and 7.0%, the latter value occurring in 2012. Total sublimation and drifting snow erosion are
- 30

relatively constant from year to year and do not show significant trends.

During 1961-1990, the small positive trends in melt ME and runoff RU were offset by a similarly small precipitation increase, resulting in an insignificant trend in SMB. This changed dramatically in the ensuing period (1991-2015), during which ME and RU trends increased six- to sevenfold. In combination with a small decrease in precipitation, this led to a sharp decrease in SMB of 3.3% per year. A synchronous increase in refreezing RF has limited the mass loss: based on annual values, the refrozen fraction RF/(RA + ME) varied between 32% and 57%, underlining the great importance of firm processes for contemporary GrIS mass balance. During 1991-2015, the refrozen fraction averaged 41%, down from 44% in 1961-1990, signalling a decrease in the retention efficiency of the GrIS firm layer. In RACMO2.3 this is mainly caused by a

5 decrease in firn pore space (Van Angelen et al., 2014), but in reality this effect is exacerbated by the formation of impenetrable ice lenses during warm summers, preventing the surface meltwater from reaching the deeper firn layers and using the full retention potential (Machguth et al., 2016).

Figure 4 clearly demonstrates the large interannual variability of the major GrIS SMB components  $P_{tot}$ , ME and RU. For ME and RU, the peak of 2012 stands out, with a modelled melt flux in excess of 1000 Gt, i.e. 1 Tt, exceeding the previous

- record of 2010 (~ 800 Gt) by a wide margin. Remarkably, the summer of 2013 saw a return to near-normal melt conditions, with melt close to the 1961-1990 average, while summer 2015 saw record melting in the northern reaches of the ice sheet (Tedesco et al., 2016). This exceptional interannual variability in the melt climate of the GrIS points towards important roles for large-scale atmospheric drivers (Fettweis et al., 2013b; Hanna et al., 2013a, 2014, 2016; McLeod and Mote, 2016) and local feedback processes. Especially important is the albedo-melt feedback (Box et al., 2012), which constitutes the darkening
- of snow once it has melted, as well as the lengthening of the exposure of dark, bare ice in the ablation zone once the winter snow has melted away (Tedesco et al., 2011). But precipitation, where feedbacks play a lesser role, is also highly variable from year to year; for instance,  $P_{tot}$  increased by ~ 300 Gt yr<sup>-1</sup> between 1971 and 1972, a year-to-year change equivalent to 40% of the long-term average. Fitting a linear trend to the standard deviation of running decadal values reveals that precipitation variability decreased by ~ 30 Gt yr<sup>-1</sup>, while that of runoff increased by approximately the same amount. The reasons for this
- are presently not clear.

#### 3.3 Spatial SMB variability

In this section we discuss the spatial distribution of changes in liquid water balance (LWB) and SMB components between the climatic period 1961-1990 and the recent period of GrIS mass loss. For a description of spatial differences in D the reader is referred to e.g. Enderlin et al. (2014) and Csatho et al. (2014). To maximise the length and to avoid spurious trends, we let the recent period start in 1991 rather than in 1995, when the first changes became noticeable. Figures 5 to 8 show the average for 1961-1990 (a) and the difference between 1991-2015 and 1961-1990 (b) of melt ME, refreezing RF, runoff RUand SMB, respectively. In these maps, mass flux (difference) is expressed as kg m<sup>-2</sup> yr<sup>-1</sup>, equivalent to mm w.e. yr<sup>-1</sup>. Note that over non-glaciated areas, RACMO2.3 uses a simpler snow model that does not calculate refreezing, only melt and runoff; SMB is therefore only calculated and physically meaningful over glaciated areas.

Figure 5a shows that over tundra, the average annual melt rate (ME) is limited by the annual snowfall, which explains the generally lower values when compared to the adjacent glaciated areas, where ablated ice is continuously replenished by glacier flow. Over glaciated areas, ME increases strongly with decreasing elevation and latitude, reaching 3500 mm w.e. yr<sup>-1</sup> in the low-lying parts of the southwestern GrIS. Although higher observed melt rates have been reported for the GrIS ablation zone, this concerns mainly locally very dark ice surfaces and/or isolated glacier tongues surrounded by ice free land that are not well resolved at the model resolution of 11 km (Fausto and van As, 2012).

Most of the meltwater produced at higher elevations refreezes in the cold firn. Figure 6a shows that refreezing (RF) peaks in the lower accumulation zone between 1000-2000 m a.s.l., where significant summer melt occurs yet the firn layer still has 5 sufficient pore space to store the meltwater. As a result, all runoff (RU) from the GrIS occurs from ice sheet regions roughly below 2000 m a.s.l. in the south and 1500 m a.s.l. in the north (Figure 7a). The resulting 1961-1990 *SMB* distribution (Figure 8a) shows relatively wide (50-150 km) ablation zones in the dry southwest, north and northeast of the GrIS, and narrow (10-50 km) ablation zones in the wetter and therefore steeper-sloped northwest and southeast of the GrIS.

- Relative to 1961-1990, melt in 1991-2015 has increased everywhere over the GrIS (Figure 5b). The change is not statistically significant on the higher ice sheet domes, where melt occurs intermittently. Integrated over the contiguous GrIS, melt has increased from 433 Gt yr<sup>-1</sup> in 1961-1990 to 581 Gt yr<sup>-1</sup> in 1991-2015, an increase of 34% (see Table 1). Not all additional liquid water reaches the ocean: part of the mass loss is buffered by increased *RF*. Table 1 and Figure 4 show that 29% of the combined increase in *ME* and *RA* is buffered by increased *RF*. Figure 6b clearly shows that this increase in *RF* is confined to areas where the firn layer has sufficient storage capacity, i.e. well above the equilibrium line. As a result, the increase in *RU*
- 15 is confined to the ablation zone and the lower accumulation zone (Figure 7b). Resulting changes in the SMB (Figure 8b) have two components: a significant decrease in the ablation zone and lower accumulation zone that mirrors the change in RU, and a partially significant increase in the interior, owing to an increase in snowfall. The result is that SMB gradients have steepened along the perimeter of the ice sheet, which is also visible in high-resolution altimetry (Helm et al., 2014).
- In line with observations, the neighbouring ice caps in the Canadian Arctic, Iceland and Svalbard have also experienced strongly increased melt rates. *ME* decreased only over some non-glaciated areas, mainly as a result of decreased (winter) snow accumulation. Only the interior parts of the highest and coldest ice caps in the northern Canadian Arctic remain free of runoff, while ice caps in the southern Canadian Arctic, Iceland and Svalbard all produce runoff from the entire ice surface.

#### 3.4 Temporal MB variability

Figure 9 combines GrIS integrated values of SMB and D into ice sheet mass balance (MB) with uncertainties as defined in

- section 2.2. Linear trends for the period 1991-2015 are indicated by dashed lines. The equivalent sea level rise (Eq. SLR) for negative MB is provided on the lower right axis. The MB values before 1996 are uncertain because reliable estimates of Dare missing, although previous work reported little difference between discharge estimates from the early 1960's and the mid 1990's (Rignot et al., 2008). Before 1995, under the assumption of constant ice discharge, we see that MB typically varied between +200 and -200 Gt yr<sup>-1</sup>, with an average close to zero. After 1995, MB becomes persistently negative, with a minimum
- in 2012 of -446  $\pm$  114 Gt yr<sup>-1</sup>, equivalent to a SLR of 1.2  $\pm$  0.3 mm yr<sup>-1</sup>. In 2013, *MB* sharply increased in response to a summer with near-normal surface melt conditions; this temporarily limited GrIS mass loss but did not eliminate it, because *D* remained elevated. After 2013, *ME* and *RU* increased once more, reducing *SMB* and increasing the mass loss to values again approaching 1 mm equivalent SLR in 2015.

The period-average mass loss we obtain here can for instance be compared to the Ice sheet Mass Balance Intercomparison Experiment (IMBIE) results in Shepherd et al. (2012), which used data of gravimetry, altimetry and MBM to reconcile differences in ice sheet mass balance. For the periods 1992-2000 and 2000-2011, average values for GrIS *MB* in that (this) study are  $-51 \pm 65$  Gt yr<sup>-1</sup> ( $-35 \pm 79$  Gt yr<sup>-1</sup>) and  $-211 \pm 37$  Gt yr<sup>-1</sup> ( $-236 \pm 86$  Gt yr<sup>-1</sup>). Although not fully independent, this good agreement suggests that the uncertainties in these numbers may actually be smaller than stated.

The decrease in MB since 1991 is significant and indicates an acceleration of the mass loss of  $16.8 \pm 2.8$  Gt yr<sup>-2</sup>. This is somewhat less than the  $21.9 \pm 1$  Gt yr<sup>-2</sup> reported by Rignot et al. (2011) for the period 1992-2010, which is obviously caused by the inclusion of years after 2012 with higher MB. For the same period 1992-2010 we obtain  $21.8 \pm 3.7$  Gt yr<sup>-2</sup>, i.e. a number very close to Rignot et al. (2011). Using trends in SMB and D as reliable indicators for mass loss partitioning, the

10 acceleration is caused for 61% by a decrease in SMB and for the remainder by an increase in D. Again owing to the inclusion of the years 2013-2015, this partitioning between the two components is somewhat closer to equality than reported in Enderlin et al. (2014).

#### 4 Discussion and conclusions

5

These results show that GrIS *MB* has been persistently negative since 1998, and continues to be negative in spite of a temporal
rebound in 2013. The mass loss that occurred each year between 2006 and 2012 was unprecedented since 1958. How significant are these recent mass losses and how do they impact the future resilience of the ice sheet?

First we note that the capacity of the firn layer to buffer runoff (Harper et al., 2012) is compromised by increased meltwater refreezing. The associated release of latent heat warms the firn, reducing its cold content and enhances dry firn compaction, which together with the refrozen mass reduces the pore space where meltwater can be stored and refrozen. Figure 10 shows

20 results of calculations with an offline firn densification model forced with RACMO2.3 output. It shows that firn temperatures in response to enhanced melt and refreezing have increased by up to 5 K in the lower accumulation area (Figure 10a). Firn air content, defined as the vertically integrated depth of the air column (expressed in m), decreased by up to 6 m in the same areas (Figure 10b). These changes are highly significant considering that typical values for total firn air content in the dry snow zone range between 20-25 m (Kuipers Munneke et al., 2015). Recent research has shown that warm summers can generate thick ice layers that prevent meltwater from reaching the deeper pores, further reducing the meltwater buffering capacity (Machguth et al., 2016).

The mass overturning rate of an ice sheet is defined as its total mass divided by the annual mass gain. We approximate the latter by the 1961-1990 average of  $SN - SU_{tot} = 632$  Gt yr<sup>-1</sup>, assuming most rainfall to fall on the lower ice sheet margins and to runoff quickly. Combined with the estimated volume of the GrIS of 2.67 x 10<sup>6</sup> km<sup>3</sup> (Vaughan et al., 2013) and an ice density

30 of 900 kg m<sup>-3</sup>, we obtain a mass overturning rate of 4,200 years. If we interpret this as the reaction time scale of an ice sheet to adjust its dynamics to changes in accumulation, it is clear that the recent changes owing to melt, with a typical timescale of decades, by far outpace a potential ice sheet thickening owing to increased snowfall in a warmer climate. In spite of that, at the current rate of mass loss, it would still take over 10,000 years to melt the entire ice sheet. Alternatively, we may state that the GrIS is significantly out of balance, noting that the average 1991-2015 mass loss (171 Gt yr<sup>-1</sup>) represents a sizeable fraction (27%) of the annual mass gain. Because D is definite positive, the situation in which SMB becomes persistently negative leads to a definite negative MB, even when the ice sheet has lost contact with the ocean, i.e. D = 0. This is therefore sometimes labelled a 'tipping point' for GrIS mass balance, beyond which the ice sheet will not be

- able to recover. At the current rate of SMB decrease (10.2  $\pm$  2.3 Gt yr<sup>-2</sup>), this tipping point would be reached between 2024 and 2043. The limited length of the time series and our incomplete knowledge of the main drivers of changes in SMB and D for now preclude firm statements about how realistic such a scenario is. It is therefore desirable that both MB components be reconstructed as far back in time as possible; this will require the smart use of climate archives, such as firn cores from the accumulation zone of the ice sheet (Box and Colgan, 2013), robust re-analysis products that cover the full 20th century (Hanna
- et al., 2011; Lee and Biasutti, 2014) and early satellite products and photogrammetry (Kjeldsen et al., 2015). Because pre-1995 values of *SMB* and *D* are similar, in this study it was not necessary to define a reference period from which cumulative anomalies are defined (Van den Broeke et al., 2009). Instead, absolute mass fluxes could simply be integrated in time and subtracted to obtain ice sheet *MB* for comparison with GRACE (Figure 2). But this pre-1995 agreement is almost certainly in part fortuitous, because uncertainties in especially *SMB*, which is a modelled quantity, and to a lesser extent in
- D, which is largely observed, remain significant. For instance, current model horizontal resolution of RACMO2.3 (11 km) is insufficient to resolve the individual, low-lying outlet glaciers of the GrIS where runoff is especially large; as a result *RU* increases when the 11 km field is statistically downscaled to 1 km resolution. This unresolved mass loss is likely in part error-compensated by snowfall in RACMO2.3 being underestimated in some regions of the ice sheet (Overly et al., 2015). In a recent study, it was moreover demonstrated that while RACMO2.3 tends to time drifting snow events well, the model likely overestimates drifting snow transport and therewith drifting snow sublimation (Lenaerts et al., 2012). This leads to uncertainties
- in SMB of 60 100 Gt yr<sup>-1</sup>, clearly dominating the uncertainty in MB (Fig. 9).

To reduce these biases and increase our diagnostic and prediction skills of GrIS mass balance, it is imperative that SMB and firn models are further improved and their horizontal resolution enhanced. This can be achieved through statistical/dynamical downscaling in combination with targeted in-situ observations. Examples of important processes that are poorly or not at all

- represented in current models are interactive snow/ice darkening by future enhanced dust/black carbon deposition or microbiological processes (Stibal et al., 2012), and sub-, supra- and englacial hydrology, including vertical and horizontal flow of meltwater in firn or over ice lenses (Machguth et al., 2016). Other emerging research topics of GrIS melt climate are the impact of atmospheric circulation changes on Greenland melt (Hanna et al., 2013a, 2014, 2016; McLeod and Mote, 2016; Tedesco et al., 2013), the impact of rain on ice sheet motion (Doyle et al., 2015), the effect of liquid water clouds on the surface energy
- balance and melt (Bennartz et al., 2013; Van Tricht et al., 2016) and the increased role of turbulent heat exchange during strong melting episodes over the margins of the GrIS (Fausto et al., 2016). Finally, it is desirable that, once developed and tested, a single, sophisticated snow model is used to simulate both the deep firn layer over the ice sheet and the seasonal snow cover over the tundra.

*Acknowledgements.* MRvdB, PKM, BN, WJvdB and BW acknowledge support of the Polar Program of the Netherlands Organisation for Scientific Research (NWO) and the Netherlands Earth System Science Centre (NESSC).

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

| SMB               | Average       | Trend       | Average        | Trend                   |
|-------------------|---------------|-------------|----------------|-------------------------|
| component         | (1961-1990)   | (1961-1990) | (1991-2015)    | (1991-2015)             |
| P <sub>tot</sub>  | $695\pm79$    | $2.1\pm1.6$ | $712\pm 64$    | $\textbf{-1.7}\pm1.8$   |
| SN                | $673\pm77$    | $1.9\pm1.6$ | $684\pm 61$    | $\textbf{-2.0} \pm 1.7$ |
| RA                | $23\pm 6$     | $0.3\pm0.1$ | $28\pm9$       | $0.3\pm0.2$             |
| SU <sub>tot</sub> | $41\pm 6$     | $0.1\pm0.1$ | $42\pm4$       | $0.0\pm0.1$             |
| $ER_{ds}$         | $1\pm 0$      | $0.0\pm0.0$ | $1\pm 0$       | $\textbf{-0.0}\pm0.0$   |
| ME                | $433\pm68$    | $1.9\pm1.4$ | $581\pm145$    | $11.4\pm3.4$            |
| RF                | $200\pm27$    | $0.9\pm0.6$ | $245\pm59$     | $3.2\pm1.5$             |
| RU                | $256\pm51$    | $1.3\pm1.1$ | $363\pm102$    | $8.4\pm2.3$             |
| SMB               | $398 \pm 112$ | $0.8\pm2.4$ | $306\pm120$    | $-10.2\pm2.3$           |
| D                 | -             | -           | $477\pm51$     | $6.6\pm0.4$             |
| MB                | -             | -           | $-171 \pm 157$ | $-16.8\pm2.8$           |

**Table 1.** Contiguous ice sheet (GrIS) averages (1961-1990, Gt yr<sup>-1</sup> with standard deviation) and trends (1961-1990 and 1991-2015, Gt yr<sup>-2</sup>, with standard error) of SMB components, discharge D and mass balance MB.