# Peer review of "On the recent contribution of the Greenland ice sheet to sea level change"

_The Cryosphere, 2016_

## Referee Comment (RC1) · Anonymous Referee #1 · 8 Jun 2016

This is an interesting and useful up-to-date analysis of recent changes in the Greenland Ice Sheet mass balance, considering independent estimates of surface mass balance, solid ice discharge and total mass change from GRACE, that is set in a longer-term context of the last 58 years. The paper is generally well-written and -presented, and the underlying analysis seems proficient, although a few aspects (detailed below) need clarification. I'd be happy to recommend full publication in TC once the authors have addressed the following points.

page 1, line 11: change to "increased refreezing prevents runoff of meltwater FROM OCCURRING, at the expense of..."

p.4, l.26: How well do the ERA-40 and ERA-I climatologies agree for the overlap period (1979-2002)? If there is any mismatch, splicing or some other adjustment of the

temperature and precipitation etc. fields may be required.

p.5, l.10: "typical uncertainties of 9% and 15% were found for ice-sheet wide integrated accumulation and ablation" - can you comment on regional-scale uncertainties, as these can be much greater and might be important in the context of this study?

p.5, l.21: who are "A. et al. (2013)"?

p.5, l.22 "The uncertainty in monthly GRACE values are..." - how are these uncertainties defined?

p.5, l.26 "we interpolated the GRACE data to the end of the month" - how? E.g. linear or sinsusoidal interpolation?

p.6, l.4: change to "are usually assumed TO BE part of..."

p.6, l.19 "Until mid-1996, cumulative D represents a straight line, because its annual value is assumed constant at the 1996 value". How reasonable is this assumption and what difference does it make to your results? Rignot et al. (2008) suggest that anomalies in D since ~1960 are quite variable in time (their Fig. 3).

p.6, l.23 re. reference to ice sheet in "approximate balance" in the mid-1990s. How do you reconcile this with Krabill et al.'s (2004) finding of a GrIS mass balance of ~60+-15 km^3 yr^-1 from 1993/4-1998/9 based on ATM data? Krabill, W., Hanna, E., Huybrechts, P., Abdalati, W., Cappelen, J., Csatho, B., Frederick, E., Manizade, S., Martin, C., Sonntag, J., Swift, R., Thomas, R. and Yungel, J. (2004). Greenland Ice Sheet: increased coastal thinning. Geophysical Research Letters, 31, L24402, doi:10.1029/2004GL021533.

p.7, l.6: ", the real uncertainties in the trends.." - again, how are these defined? Also, comma at beginning of the words in (my above) parentheses should be a semicolon.

p.7, l.31: change to "preventing the surface meltwater FROM REACHING the deeper firn layers and USING the full retention potential".

[Figure]

p.8, l.2 suggest modify/add text in CAPS as follows; "Remarkably, the summer of 2013 saw a return to near-normal melt conditions, with melt close to the 1961-1990 average, WHILE SUMMER 2015 SAW RECORD MELTING IN THE NORTHERN REACHES OF THE ICE SHEET (TEDESCO ET AL 2016). This exceptional interannual variability in the melt climate of the GrIS points towards important roles for large-scale atmospheric drivers (Fettweis et al. 2013, HANNA ET AL. 2013, 2014 & 2016, MCLEOD & MOTE 2016) and local feedback processes.

And add extra references:

Hanna, E., Jones, J. M., Cappelen, J., Mernild, S. H., Wood, L., Steffen, K. and Huy-brechts, P. (2013), The influence of North Atlantic atmospheric and oceanic forcing effects on 1900–2010 Greenland summer climate and ice melt/runoff. Int. J. Climatol., 33: 862–880. doi: 10.1002/joc.3475

Hanna, E., Fettweis, X., Mernild, S. H., Cappelen, J., Ribergaard, M. H., Shuman, C. A., Steffen, K., Wood, L. and Mote, T. L. (2014), Atmospheric and oceanic climate forcing of the exceptional Greenland ice sheet surface melt in summer 2012. Int. J. Climatol., 34: 1022–1037. doi: 10.1002/joc.3743

Hanna, E., Cropper, T. E., Hall, R. J. and Cappelen, J. (2016), Greenland Block-ing Index 1851–2015: a regional climate change signal. Int. J. Climatol.. doi: 10.1002/joc.4673

McLeod, J. T. and Mote, T. L. (2016), Linking interannual variability in extreme Green-land blocking episodes to the recent increase in summer melting across the Greenland ice sheet. Int. J. Climatol., 36: 1484–1499. doi: 10.1002/joc.4440

Tedesco, M., T. Mote, X. Fettweis, E. Hanna, J. Jeyaratnam, J.F. Booth, R. Datta, K. Briggs (2016) Arctic cutoff high drives the poleward shift of a new Greenland melting record. Nature Climate Change, in press.

p.8, l.6 Has year-to-year variability of annual precipitation decreased significantly for
the whole period? The graph suggests there may have been a decrease and maybe an opposite trend (recently increased variability) for runoff - ?

p.9, l.13 "previous work reported little difference between discharge estimates from the early 1960s and the mid 1990s (Rignot et al. (2008) - this is strictly correct but Rignot et al. (2008, their Fig. 3) also suggest a 50-100 Gt yr^-1 change in discharge between these two periods and the intermediate mid-late 1970s and early 1980s period.

p.10, l.30 "use of...robust re-analysis products that cover the full 20th century" - add Hanna et al. (2011) reference to Lee & Biasutti (2014): Hanna, E., Huybrechts, P., Cappelen, J., Steffen, K., Bales, R.C., Burgess, E.W., McConnell, J.R., Steffensen, J. P., van den Broeke, M., Wake, L., Bigg, G.R., Griffiths, M. and Savas, D. (2011). Greenland Ice Sheet surface mass balance 1870 to 2010 based on Twentieth Century Reanalysis, and links with global climate forcing. Journal of Geophysical Research - Atmospheres, 116, D24121, doi:10.1029/2011JD016387.

p.11, l.14: add "impact of atmospheric circulation changes on Greenland melt" to "Other emerging research topics of GrIS melt climate.", and add the following references: Tedesco et al. 2013, Hanna et al. 2013, 2014 & 2016, McLeod & Mote 2015. See above for reference details except for: Tedesco, M., Fettweis, X., Mote, T., Wahr, J., Alexander, P., Box, J. E., and Wouters, B.: (2013) Evidence and analysis of 2012 Greenland records from spaceborne observations, a regional climate model and re-analysis data, The Cryosphere, 7, 615-630, doi:10.5194/tc-7-615-2013.

---

## Referee Comment (RC2) · X. Fettweis (Referee) · 24 Jun 2016

X. Fettweis (Referee)

xavier.fettweis@ulg.ac.be

This paper presents an update of the Rignot's papers evaluating the recent SMB and iceberg discharge changes over the Greenland ice sheet using "state of the art" tools. It is not really innovative but this analysis covers the last years not included in the Rignot's papers and explains and analyses the recent changes in SMB. The discussion over the mass overturning rate is also particularly interesting. Therefore, it certainly deserves to be accepted for publication in TC. The paper is very well written and is ready to be published like this. I support most of the remakes from reviewer #1 and in particular the impact of using inhomogeneous forcings (ERA-40 vs ERA-Int) on the RACMO outputs.

As explained in Fetweis et al. (2013), MAR forced by ERA-40 (over 1980-1999) over-estimates precipitation (+30 GT/yr $\sim$ +5%) in respect to MAR forced by ERA-Interim

because the ERA-40 based high atmosphere is too wet than ERA-Interim as a result of biases found in the ERA-40 humidity scheme and corrected afterwards in ERA-Interim. However, this anomaly is homogeneous over the whole integration domain explaining why there are not locally significant discrepancies between both forcing. I think that these 5 % are included in the error bar used here. Therefore, it should be good to add a sentence pointing to this issue in the text.

ref: Fettweis, X., Franco, B., Tedesco, M., van Angelen, J. H., Lenaerts, J. T. M., van den Broeke, M. R., and Gallée, H.: Estimating the Greenland ice sheet surface mass balance contribution to future sea level rise using the regional atmospheric climate model MAR, The Cryosphere, 7, 469-489, doi:10.5194/tc-7-469-2013, 2013.

---

## Author Comment (AC1) · 10 Aug 2016

The comment was uploaded in the form of a supplement:
http://www.the-cryosphere-discuss.net/tc-2016-123/tc-2016-123-AC1-supplement.pdf

---

## Author Response (AR1)

**TC-2016-123 "On the recent contribution of the Greenland ice sheet to sea level change"**

**Response to Reviewers**

We are grateful for the comments provided by the reviewers. Please find below our answers (in red) to the reviewer's comments (in blue) and the suggested changes in the MS main text (*red, italic*).

**Anonymous Referee #1**

This is an interesting and useful up-to-date analysis of recent changes in the Greenland Ice Sheet mass balance, considering independent estimates of surface mass balance, solid ice discharge and total mass change from GRACE, that is set in a longer-term context of the last 58 years. The paper is generally well-written and -presented, and the underlying analysis seems proficient, although a few aspects (detailed below) need clarification. I'd be happy to recommend full publication in TC once the authors have addressed the following points.

page 1, line 11: change to "increased refreezing prevents runoff of meltwater FROM OCCURRING, at the expense of..."

Changed as suggested.

p.4, l.26: How well do the ERA-40 and ERA-I climatologies agree for the overlap period (1979-2002)? If there is any mismatch, splicing or some other adjustment of the temperature and precipitation etc. fields may be required.

Here we follow the suggestion of reviewer 2, and added:

*In previous versions of RACMO2, the impact of using inhomogeneous forcing before and after 1979 (ERA-40 vs ERA-INT) was found to be small (Ettema et al., 2009). According to Fettweis et al. (2013), the regional climate model MAR shows precipitation to be 5% greater when forced by ERA-40 compared to ERA-Interim for the period 1980-1999, probably a result of biases in the ERA-40 humidity scheme that were corrected in ERA-Interim. This uncertainty falls within the error bar used here.*

We also added the reference:

*Fettweis, X., Franco, B., Tedesco, M., van Angelen, J. H., Lenaerts, J. T. M., van den Broeke, M. R. and Gallée, H., 2013: Estimating the Greenland ice sheet surface mass balance contribution to future sea level rise using the regional atmospheric climate model MAR, The Cryosphere, 7, 469-489, doi:10.5194/tc-7-469-2013.*

p.5, l.10: "typical uncertainties of 9% and 15% were found for ice-sheet wide integrated accumulation and ablation" - can you comment on regional-scale uncertainties, as these can be much greater and might be important in the context of this study?

Although we have attempted this, the scarcity of accumulation and ablation measurements does not allow for a regionalization of the error. To clarify this, we added:

*The scarcity of accumulation and ablation measurements does not allow for a further regionalization of the uncertainty, but obviously uncertainties can be significantly larger for smaller areas and sub-climatological time periods.*

p.5, l.21: who are "A. et al. (2013)"?

*These are Geruo A and co-authors (https://www.ess.uci.edu/people/geruoa).*

p.5, l.22 "The uncertainty in monthly GRACE values are..." - how are these uncertainties defined?

To explain, we added:

*For the monthly values, the uncertainty is computed by conservatively assuming that all observed signals with a periodicity smaller than 6 months are due to noise (Wahr et al., 2006). For the trends, the quoted uncertainty takes into account methodological differences in the processing of the level-1 GRACE data, the uncertainty in the GIA correction, the formal error of the least-square fit and aliasing of ocean tides (Wouters at al., 2013).*

We added two references:

*Wahr, J., S. Swenson, and I. Velicogna, 2006: Accuracy of GRACE mass estimates, Geophys. Res. Lett., 33, L06401, doi:10.1029/2005GL025305.*

*Wouters, N., J. L. Bamber, M. R. van den Broeke, J. T. M. Lenaerts and I. Sasgen, 2013: Limits in detecting acceleration of ice sheet mass loss due to climate variability, Nature Geoscience 6, doi: 10.1038/ngeo1874.*

p.5, l.26 "we interpolated the GRACE data to the end of the month" - how? E.g. linear or sinsusoidal interpolation?

We used linear interpolation. To clarify, the text is changed to:

*"we linearly interpolated the GRACE data to the end of the month"*

p.6, l.4: change to "are usually assumed TO BE part of..."

Changed as suggested.

p.6, l.19 "Until mid-1996, cumulative D represents a straight line, because its annual value is assumed constant at the 1996 value". How reasonable is this assumption and what difference does it make to your results? Rignot et al. (2008) suggest that anomalies in D since ~1960 are quite variable in time (their Fig. 3).

No discharge estimates are available between 1960 and 1996. In Fig. 3 of Rignot and others (2008) total mass balance is not observed but reconstructed by assuming D to depend linearly on del_SMB (see their Fig. 2). The reviewer is correct that our results would change if we have time varying estimates of D before 1996. However, in the case D would differ significantly from average SMB we would have adopted the anomaly method as in Van den Broeke and others (2009), as explained in the text.

p.6, l.23 re. reference to ice sheet in "approximate balance" in the mid-1990s. How do you reconcile this with Krabill et al.'s (2004) finding of a GrIS mass balance of ~60+-15 km^3 yr^-1 from 1993/4-1998/9 based on ATM data?

Krabill, W., Hanna, E., Huybrechts, P., Abdalati, W., Cappelen, J., Csatho, B., Frederick, E., Manizade, S., Martin, C., Sonntag, J., Swift, R., Thomas, R. and Yungel, J. (2004). Greenland Ice Sheet: increased coastal thinning. Geophysical Research Letters, 31, L24402, doi:10.1029/2004GL021533.

As a result of SMB variability, large year-to-year variations are possible in MB. Our statement is based on MB compilation studies such as Hanna and others (Nature, 2013) and Briggs and others (2016, EOS), which suggest that MB was essentially close to zero in that epoch. These compilations do not show studies where MB was significantly positive, i.e. they discarded the Krabill result. We have added the Hanna and others (2013) reference.

p.7, l.6: ", the real uncertainties in the trends.." - again, how are these defined? Also, comma at beginning of the words in (my above) parentheses should be a semicolon.

Semicolon has been inserted. For uncertainty in the GRACE trend, see added explanation above for p. 5, l. 22. For SMB-D, this uncertainty is derived following Van den Broeke and others (2009). We have added the following sentence:

*For the trend in cumulative SMB-D, an uncertainty is derived following Van den Broeke and others (2009).*

p.7, l.31: change to "preventing the surface meltwater FROM REACHING the deeper firn layers and USING the full retention potential".

Changed as suggested.

p.8, l.2 suggest modify/add text in CAPS as follows; "Remarkably, the summer of 2013 saw a return to near-normal melt conditions, with melt close to the 1961-1990 average, WHILE SUMMER 2015 SAW RECORD MELTING IN THE NORTHERN REACHES OF THE ICE SHEET (TEDESCO ET AL 2016). This exceptional interannual variability in the melt climate of the GrIS points towards important roles for large-scale atmospheric drivers (Fettweis et al. 2013, HANNA ET AL. 2013, 2014 & 2016, MCLEOD & MOTE 2016) and local feedback processes. And add extra references:

Hanna, E., Jones, J. M., Cappelen, J., Mernild, S. H., Wood, L., Steffen, K. and Huybrechts, P. (2013), The influence of North Atlantic atmospheric and oceanic forcing effects on 1900–2010 Greenland summer climate and ice melt/runoff. Int. J. Climatol., 33: 862–880. doi: 10.1002/joc.3475

Hanna, E., Fettweis, X., Mernild, S. H., Cappelen, J., Ribergaard, M. H., Shuman, C. A., Steffen, K., Wood, L. and Mote, T. L. (2014), Atmospheric and oceanic climate forcing of the exceptional Greenland ice sheet surface melt in summer 2012. Int. J. Climatol., 34: 1022–1037. doi: 10.1002/joc.3743

Hanna, E., Cropper, T. E., Hall, R. J. and Cappelen, J. (2016), Greenland Blocking Index 1851–2015: a regional climate change signal. Int. J. Climatol.. doi: 10.1002/joc.4673

McLeod, J. T. and Mote, T. L. (2016), Linking interannual variability in extreme Greenland blocking episodes to the recent increase in summer melting across the Greenland ice sheet. Int. J. Climatol., 36: 1484–1499. doi: 10.1002/joc.

Tedesco, M., T. Mote, X. Fettweis, E. Hanna, J. Jeyaratnam, J.F. Booth, R. Datta, K. Briggs (2016) Arctic cutoff high drives the poleward shift of a new Greenland melting record. Nature Climate Change, in press.

Changed as suggested.

p.8, l.6 Has year-to-year variability of annual precipitation decreased significantly for the whole period? The graph suggests there may have been a decrease and maybe an opposite trend (recently increased variability) for runoff - ?

Interesting point; the running decadal standard deviation indeed decreased for precipitation and increased for runoff. We added the following sentence:

*Fitting a linear trend to the standard deviation of running decadal values reveals that precipitation variability decreased by ~30 Gt yr-1, while that of runoff increased by approximately the same amount. The reasons for this are presently not clear.*

p.9, l.13 "previous work reported little difference between discharge estimates from the early 1960s and the mid 1990s (Rignot et al. (2008) - this is strictly correct but Rignot et al. (2008, their Fig. 3) also suggest a 50-100 Gt yrˆ-1 change in discharge between these two periods and the intermediate mid-late 1970s and early 1980s period.

See previous answer to p. 6, l. 19.

p.10, l.30 "use of...robust re-analysis products that cover the full 20th century" - add Hanna et al. (2011) reference to Lee & Biasutti (2014):

Hanna, E., Huybrechts, P., Cappelen, J., Steffen, K., Bales, R.C., Burgess, E.W., McConnell, J.R., Steffensen, J. P., van den Broeke, M., Wake, L., Bigg, G.R., Griffiths, M. and Savas, D. (2011). Greenland Ice Sheet surface mass balance 1870 to 2010 based on Twentieth Century Reanalysis, and links with global climate forcing. Journal of Geophysical Research - Atmospheres, 116, D24121, doi:10.1029/2011JD016387.

Changed as suggested.

p.11, l.14: add "impact of atmospheric circulation changes on Greenland melt" to "Other emerging research topics of GrIS melt climate.", and add the following references: Tedesco et al. 2013, Hanna et al. 2013, 2014 & 2016, McLeod & Mote 2015. See above for reference details except for:

Tedesco, M., Fettweis, X., Mote, T., Wahr, J., Alexander, P., Box, J. E., and Wouters, B.: (2013) Evidence and analysis of 2012 Greenland records from spaceborne observations, a regional climate model and reanalysis data, The Cryosphere, 7, 615-630, doi:10.5194/tc-7-615-2013.

Changed as suggested.

**Referee 2 (Xavier Fettweis)**

This paper presents an update of the Rignot's papers evaluating the recent SMB and iceberg discharge changes over the Greenland ice sheet using "state of the art" tools. It is not really innovative but this analysis covers the last years not included in the Rignot's papers and explains and analyses the recent changes in SMB. The discussion over the mass overturning rate is also particularly interesting. Therefore, it certainly deserves to be accepted for publication in TC. The paper is very well written and is ready to be published like this. I support most of the remakes from reviewer #1 and in particular the impact of using inhomogeneous forcings (ERA-40 vs ERA-Int) on the RACMO outputs.

As explained in Fetweis et al. (2013), MAR forced by ERA-40 (over 1980-1999) overestimates precipitation (+30 GT/yr ~ +5%) in respect to MAR forced by ERA-Interim C1 TCD Interactive comment Printer-friendly version Discussion paper because the ERA-40 based high atmosphere is too wet than ERA-Interim as a result of biases found in the ERA-40 humidity scheme and corrected afterwards in ERA-Interim. However, this anomaly is homogeneous over the whole integration domain explaining why there are not locally significant discrepancies between both forcing. I think that these 5 % are included in the error bar used here. Therefore, it should be good to add a sentence pointing to this issue in the text.

ref: Fettweis, X., Franco, B., Tedesco, M., van Angelen, J. H., Lenaerts, J. T. M., van den Broeke, M. R., and Gallée, H.: Estimating the Greenland ice sheet surface mass balance contribution to future sea level rise using the regional atmospheric climate model MAR, The Cryosphere, 7, 469-489, doi:10.5194/tc-7-469-2013, 2013

We follow the suggestion and added:

*In previous versions of RACMO2, the impact of using inhomogeneous forcing before and after 1979 (ERA-40 vs ERA-Int) was found to be small (Ettema et al., 2009). According to Fettweis et al. (2013), the regional climate model MAR shows precipitation to be 5% greater when forced by ERA-40 compared to ERA-Interim for the period 1980-1999, probably a result of biases in the ERA-40 humidity scheme that were corrected in ERA-Interim. This uncertainty falls within the error bar used here.*

We also added the reference:

*Fettweis, X., Franco, B., Tedesco, M., van Angelen, J. H., Lenaerts, J. T. M., van den Broeke, M. R. and Gallée, H., 2013: Estimating the Greenland ice sheet surface mass balance contribution to future sea level rise using the regional atmospheric climate model MAR, The Cryosphere, 7, 469-489, doi:10.5194/tc-7-469-2013.*